# EMERGENT COLLECTIVE INTELLIGENCE FROM MASSIVE-AGENT COOPERATION AND COMPETITION

## ABSTRACT

Inspired by organisms evolving through cooperation and competition between different populations on Earth, we study the emergence of artificial collective intelligence through massive-agent reinforcement learning. To this end, We propose a new massive-agent reinforcement learning environment, Lux, where dynamic and massive agents in two teams scramble for limited resources and fight off the darkness. In Lux, we build our agents through the standard reinforcement learning algorithm in curriculum learning phases and leverage centralized control via a pixel-to-pixel policy network. As agents co-evolve through self-play, we observe several stages of intelligence, from the acquisition of atomic skills to the development of group strategies. Since these learned group strategies arise from individual decisions without an explicit coordination mechanism, we claim that artificial collective intelligence emerges from massive-agent cooperation and competition. We further analyze the emergence of various learned strategies through metrics and ablation studies, aiming to provide insights for reinforcement learning implementations in massive-agent environments.

## 1 INTRODUCTION

Complex group and social behaviors widely exist in humans and animals on Earth. In a vast ecosystem, the simultaneous cooperation and competition between populations and the changing environment serve as a natural driving force for the co-evolution of massive numbers of organisms (Wolpert & Tumer, 1999; Dawkins & Krebs, 1979). This large-scale co-evolution between populations has enabled group strategies for tasks individuals cannot accomplish (Ha & Tang, 2022). Inspired by this self-organizing mechanism in nature, i.e., collective intelligence emerges from massive-agent cooperation and competition, we propose to simulate the emergence of collective intelligence through training reinforcement learning agents in a massive-agent environment. We hope this can become a stepping stone toward massive-agent reinforcement learning research and an inspiring method for complex massive-agent problems.

Recent progress in multi-agent reinforcement learning (MARL) demonstrates its potential to complete complex tasks through multi-agent cooperation, such as playing StarCraft2 (Vinyals et al., 2019) and DOTA2 (Berner et al., 2019). However, the number of agents is still limited to dozens in those scenarios, far away from natural populations. To support large-scale multi-agent cooperation and competition, we reintroduce the massive-agent setting into multi-agent reinforcement learning. To this end, we propose Lux, a cooperative and competitive environment where hundreds of agents in two populations scramble for limited resources and fight off the darkness. We believe Lux is a suitable testbench for experimenting with collective intelligence because it provides an open environment for hundreds of agents to cooperate, compete and evolve.

From the algorithmic perspective, the massive-agent setting poses great difficulties to reinforcement learning algorithms since the credit assignment problem becomes increasingly challenging. Some research (Lowe et al., 2017) focuses on the credit assignment problem between multi-agents, however, it lacks the scalability to massive-agent scenarios. To overcome that, we present a centralized control solution for Lux using a pixel-to-pixel modeling architecture (Han et al., 2019) coupled with Proximal Policy Optimization (PPO) (Schulman et al., 2017) algorithm. Using that solution, we avoid the problem of credit assignment, with up to a 90% win rate versus the state-of-the-art policy

(Isaiah et al., 2021) proposed by the Toad Brigade team (TB) which won first place in the Lux AI competition on Kaggle [1].

Through self-play and curriculum learning phases, we observe several stages of the massive-agent co-evolution, from atomic skills such as moving and building to group strategies such as efficient territory occupation and long-term resource management. Note that group strategies arise from individual decisions without any explicit coordination mechanism or hierarchy, demonstrating how collective intelligence arises with co-evolution. Through quantitative analyses, further evidence shows that collective intelligence can emerge from massive-agent cooperation and competition, leading to behaviors beyond our expectations. For example, agents learn to stand in a diagonal row and move as a whole to segment off parts of the map as shown in Figure 1. Without any prior knowledge, this efficient strategy emerges from spontaneous exploration. Furthermore, we perform a detailed ablation study to illustrate some implementation techniques which may be helpful in massive-agent reinforcement learning.

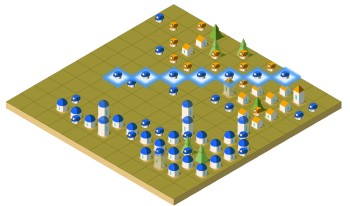 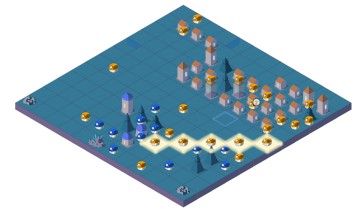

(a) Blue is our policy and Yellow is TB.   (b) Yellow is our policy and Blue is TB.

Figure 1: Two episodes between our policy and TB where our *Workers* stand in a diagonal row. Our agents discover it as an efficient way to expand the territory and limit the enemy's movement.

Our main contributions are 1) we reintroduce massive-agent reinforcement learning as a scenario for studying collective intelligence and propose a new environment, Lux, as a starting point. 2) we provide evidence that collective intelligence emerges from co-evolution through massive agents' cooperation and competition in Lux. 3) we discuss the implementation details of our solution, which may provide valuable insights into massive-agent reinforcement learning.

## 2 RELATED WORK

**Multi-Agent Environments.** Many environments such as Multi-agent Particle Environment (MPE) (Lowe et al., 2017) and Google Research Football (Kurach et al., 2020) are proposed to study multi-agent cooperation and competition. For multi-agent cooperation, StarCraft Multi-Agent Challenge (SMAC) (Samvelyan et al., 2019) provides a common testbench. However, SMAC focuses on decentralized micromanagement scenarios with only approximately 30 agents in play. In massive-agent environments, Neural MMO (Suarez et al., 2021) is an open-ended Massively Multi-player Online (MMO) game environment with up to 1024 agents. MAgent (Zheng et al., 2018) is a grid world environment that supports up to a million agents. We propose Lux, a massive-agent reinforcement learning environment, which can support thousands of agents simultaneously acting at one step. Unlike previous massive-agent environments, Lux incorporates Real-Time-Strategy (RTS) game dynamics that are similar to Battlecode (2022) and MiniRTS (Tian et al., 2017). Moreover, Lux scales up the number of agents with frequent spawns and deaths, which opens up the potential for complex strategies in such a large-scale and highly dynamic scenario.

**Credit Assignment in MARL.** Credit assignment between agents (Chang et al., 2003) is a crucial challenge in multi-agent cooperation. Several value-based multi-agent algorithms (Sunehag et al., 2017; Rashid et al., 2018; Iqbal & Sha, 2018)) decompose global value into individual values using a linear model or neural network, which can be viewed as an implicit way of credit assignment. Another way of doing this is computing an agent-specific advantage function. For example, Foerster et al. (2017) uses counterfactual regret to measure contributions to the team. In complex games, Berner et al. (2019) and Ye et al. (2020) use hand-crafted team-based rewards for each agent as an explicit method of credit assignment. Compared to the implicit value decomposition method, this

---

[1]https://www.kaggle.com/c/lux-ai-2021/

explicit reward-shaping method requires prior domain knowledge and lacks generalization ability. However, both of them are limited to small population scenarios and are hard to scale to massive and dynamic agents. Han et al. (2019) handles this problem using grid-wise centralized policy instead of decentralized policy. It uses a convolutional neural network to map from pixel-wise observations to actions over each pixel, which avoids the credit assignment problem while achieving efficient multi-agent collaboration. Following this, we adapt this pixel-to-pixel architecture to the Lux environment with the PPO algorithm (Schulman et al., 2017) and curriculum learning phases.

**Collective Intelligence and Emergence Behaviors.** Collective Intelligence, including self-organization and emergent behaviors (Wolpert & Tumer, 1999; Woolley et al., 2010), has a long history connected with biological and economic studies. Research on emergent behavior usually emphasizes that group strategies emerge from multi-agent co-evolution in a designed environment rather than hand-crafted collaboration mechanisms. Baker et al. (2019) uses reinforcement learning agents and autocurricula (Leibo et al., 2019) in the Hide-and-seek environment, leading to the emergence of tool use. Yang et al. (2018) proposes using million-agent reinforcement learning to study how the agents' grouping behaviors will change with the environmental resources. Zheng et al. (2021) uses a two-level, deep RL framework to train agents and a social planner in an economic environment, where optimal taxation policy emerges as the result of co-adaptation. Johanson et al. (2022) studies the emergence of bartering behavior in a microeconomics-based environment with producers and consumers. Dynamics in those environments usually induce agents' behaviors within human comprehension, thus limiting the possible emergent strategies. Since RTS games provide a perfect Petri dish for collective intelligence, our study absorbs RTS game dynamics into the environment where simple rules may induce complex group strategies.

## 3 LUX

Like the Earth, a suitable environment for collective intelligence to evolve must support massive agents' competition and cooperation. For that purpose, we propose an open-sourced environment Lux, where hundreds of agents in two teams compete for resources and build cities as illustrated in Figure 2.

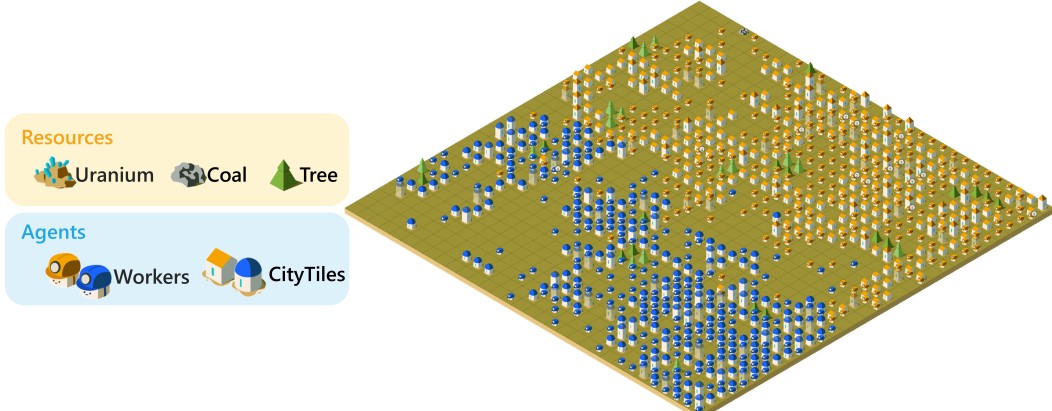

Figure 2: **A snapshot of Lux with hundreds of agents in two teams.** *Workers* can collect resources and build *CityTiles*. At night, *CityTiles* and *Workers* need fuel to stay alive and will be consumed by darkness if fuel runs out. The team that owns more cities wins in the end.

**Setups.** The map is a 2D square grid of size 12 to 32, scattered with different resources. An episode consists of 360 turns, split into 9 Day/Night cycles of 30 days and 10 nights. There are two basic units named *Worker* and *CityTile*. Each team starts with one *Worker* and one *CityTile*. *Workers* can collect resources and build *CityTiles* and *CityTiles* can also build new *Workers*. *Workers* collect adjacent resources automatically and convert resources to fuel when standing upon a friendly *CityTile*. At night, *CityTiles* and *Workers* consume fuel to stay alive and will be consumed by darkness if the fuel runs out. At the end of the game, the team with more *CityTiles* wins. More details about the environment are in Appendix A.1.

**Observation and Action Space.** For observation, each team has perfect information about the game state, including the global map, its own, and the opponent's information. For actions, each team needs to make decisions for every *Worker* and *CityTile*. A *Worker* can move in 4 cardinal directions and build a *CityTile* when it has enough resources. *Workers* however cannot move onto a tile with an enemy *CityTile* or *Worker*. A *CityTile* can build a *Worker* or research to increase the team's research points. Sufficient research points unlock the ability for the team's *Workers* to mine high-level resources that convert to more fuel like coal and uranium.

**MARL in Lux.** For MARL research, Lux raises a challenging situation for multi-agent modeling and the credit assignment problem. Distinguished from other environments, the number of agents in Lux is massive and dynamic. For a $32 \times 32$ map, the number of agents in a team can rise to $1000$. Moreover, *Workers* and *CityTiles* are built and lost all the time, bringing difficulty for multi-agent modeling of dynamic agents. A carefully-designed credit assignment scheme may be useful in small-scale problems; however, with massive and dynamic agents, it becomes impractical due to the combinatorial complexity. Furthermore, the win-or-lose sparse reward throws another challenge on the hard exploration.

**RTS in Lux**. At first sight, Lux seems like a pocket-sized RTS game like StarCraft2. Agents in Lux need to balance economic decisions and individual control, which requires high-level coordination between hundreds of agents. However, the major difference between Lux and RTS games is the way of controlling units. In RTS games, the low-level unit actions are executed by fixed rules, which allows human players or AI to focus on macro-strategies and economic decisions. In Lux, atomic actions such as moving and building are all controlled by the learned policy, resulting in an action space of approximately $10^{180}$, magnitudes beyond StarCraft2 (Vinyals et al., 2019). Thus, a successful policy needs to learn atomic skills and group strategies together, which is significant in the emergence of collective intelligence.

## 4 METHODOLOGY

Overall, our policy is trained using the standard algorithm PPO (Schulman et al., 2017) with Generalized Advantage Estimation (GAE) (Schulman et al., 2016). For massive-agent coordination, we use a pixel-to-pixel architecture as the centralized policy (Han et al., 2019; Isaiah et al., 2021), taking both observations and actions as images and using the ResNet (He et al., 2016) structure as the backbone. To address the sparse reward problem, we design three phases with different rewards as a progressive curriculum. For clarity, we refer to the "agent" as each unit on the map and refer to the "policy" as the centralized policy network that controls every agent.

### 4.1 PIXEL-TO-PIXEL ARCHITECTURE

We model the massive-agent control problem using a centralized policy with a pixel-to-pixel architecture (Han et al., 2019). The policy network takes images as input observations and outputs actions over each pixel in the form of an action map. More implementation details are in Appendix A.2.

**Policy Network Architecture.** The architecture of our policy network is pictured in Figure 3. The input image ($C \times H \times W$) consists of $C$ channels containing information about itself, the opponent, and the global state. We use a ResNet-style convolutional network as the backbone. For actions, we use a convolutional layer with kernel size 1 and output channels as the action dimensions. Moreover, we use a flattened layer and a fully-connected layer for the value estimation as the critic. A valid action mask is used to eliminate unnecessary exploration.

**Why Centralized Policy.** In Lux, our policy needs to control hundreds of agents each time step. While the decentralized policy in MARL is computationally efficient and easy to scale, it needs a carefully-designed credit assignment mechanism. In Lux, however, as agents are massive and dynamic, the credit assignment problem becomes increasingly challenging. To avoid that, we adopt a centralized policy controlling every agent over the map. This pixel-to-pixel architecture with a convolutional network leverages the advantage of centralized and decentralized methods. Convolutional layers work as a parameter-sharing mechanism across agents, similar to shared policy networks in decentralized methods. This parameter-sharing mechanism improves learning efficiency via data reuse. Furthermore, the deep stacked structure provides a large receptive field for global information

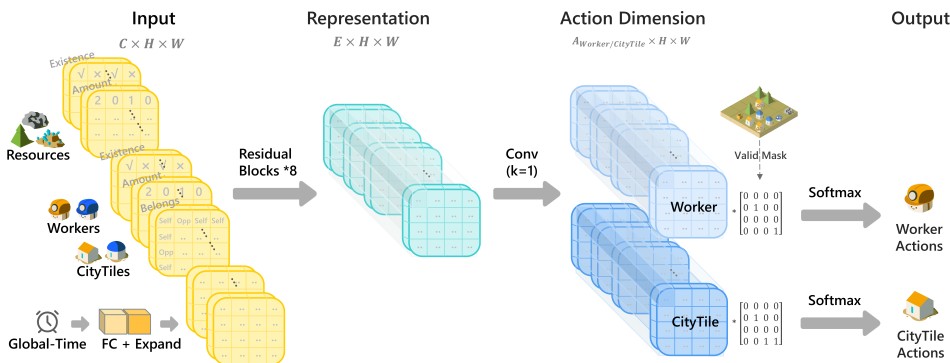

Figure 3: **Policy network architecture.** $C$ is the input channels and $H, W$ denote the map height and width. $E$ is the feature map channel through the backbone. Output channels $A_{\text{Worker/CityTile}}$ are the corresponding action dimensions.

extraction and multi-agent communication, which naturally avoids the trouble of credit assignment (Han et al., 2019).

## 4.2 CURRICULUM TRAINING PHASES

The objective of agents in Lux is to own more *CityTiles* than the opponent, but the final result only provides a sparse reward (1 for win, $-1$ for lose), resulting in the hard exploration problem (Badia et al., 2020). Reward shaping is a common method to handle this problem in reinforcement learning. However, hand-crafted rewards can easily direct agents into specific behavioral patterns with limited strategies. Hence, we design three phases with different rewards as a progressive curriculum. First, we use a dense reward to guide the policy towards basic skills. Then we gradually reduce the learning signals and utilize the sparse reward to encourage the policy to explore more diversified strategies.

**Phase 1: Dense Rewards for Basic Skills.** At first, we use hand-crafted dense rewards to encourage basic skills. Specifically, four kinds of behaviors are given rewards, namely, the increase of *Workers* and *CityTiles*, *Research Points* and fuel. More details are in Appendix A.2.

**Phase 2: Sparse Reward with Scaled Signals.** In Phase 2, a reward is given only when an episode ends. However, our policy still needs guidance through long-term reasoning. We modify the reward with a slight signal about the win condition, i.e., $\pm\sqrt{|N_{self} - N_{op}|}$, where $N_{self}$ and $N_{op}$ denote the number of our own and the enemy's *CityTiles*, encouraging to own more *CityTiles* for the win.

**Phase 3: Win-or-Lose Sparse Reward.** The win-or-lose sparse reward (1 for win and $-1$ for lose) is applied in the final phase. After human-designed guidance in the first two phases, the win-or-lose sparse reward encourages our policy to explore more advanced strategies.

## 5 EMERGENT COLLECTIVE INTELLIGENCE

Through massive-agent cooperation and competition, we have observed three stages of our agents' evolution. Training from scratch, agents quickly acquire atomic skills such as collecting resources and building cities. After around 5 million episodes, an elementary level of coordination appears on the regional scale with dozens of agents. As training proceeds, the coordination expands from regional to global scope, which includes long-term economic decisions and precise control of hundreds of agents. Those global strategies naturally arise from individual decisions due to massive-agent interaction and co-evolution without any explicit coordination mechanism, signifying the emergence of collective intelligence.

### 5.1 ATOMIC SKILLS

The first step of our agents is to get a grasp of atomic skills. Guided by dense rewards, *Workers* learn to move toward resources to collect fuel, and build and fuel the *CityTiles*, as shown in Figure

4a. However, at this stage agents are more likely to work alone and unable to make group decisions. For example, *Workers* tend to build more *CityTiles* than they can support, leading to a sudden loss of large cities as illustrated in Figure 4b as they run out of fuel.

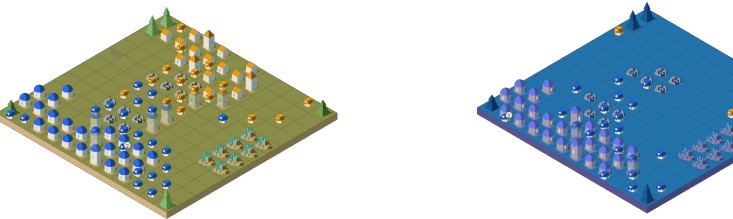

(a) *Workers* collect resources and build *CityTiles*.        (b) *CityTiles* run out of fuel and collapse.

Figure 4: **Illustration of atomic skills in a self-play episode.** Agents acquire atomic skills such as collecting resources and building *CityTiles*. However, due to a lack of group coordination, *CityTiles* often burn out fuel and collapse.

## 5.2    REGIONAL COORDINATION

As training proceeds, regional coordination appears, which involves dozens of agents in a local area. For example, agents learn to carefully choose locations before building a *CityTile* and develop self-organizing patterns for occupying resources efficiently. We describe a few examples of regional strategies:

**Construction Planning.** As *CityTiles* built next to each other can share fuel and reduce cost at night, agents gradually learn that the locations of *CityTiles* are important in city survival and fuel saving. We find that agents discover several patterns of construction planning as visualized in Figure 5: 1) build *CityTiles* near the resources for quicker access to fuel sources. 2) build *CityTiles* in a long row to form cities that act like the Great Wall to prevent enemies' aggression. 3) build *CityTiles* in blocks to reduce fuel costs at night.

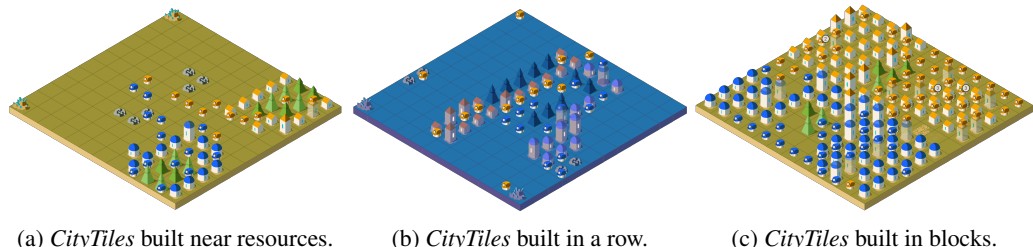

(a) *CityTiles* built near resources.        (b) *CityTiles* built in a row.        (c) *CityTiles* built in blocks.

Figure 5: **Three emergent patterns of construction planning.** a) build near resources for quick access to fuel. b) build in a row as the Great Wall for defense. c) build in blocks to save fuel.

We use the city survival ratio (the final number of *CityTiles* divided by the most number of *CityTiles* in one episode) to measure how these building patterns work quantitatively. As shown in Figure 6a, the regional-scale construction planning effectively helps *CityTiles* fight off the darkness.

**Territory Division.** We have also observed a self-organizing structure where several *Workers* stand in a diagonal row shown in Figure 1. Those *Workers* simultaneously move forward and backward as a whole to keep the formation, and when any of them die, a new *Worker* nearby will fill in. In this shape, they can effectively guard and expand the team's territory and limit the enemy's movement. We measure a statistic called Five-Diagonal (how many times five or more *Workers* stand in a diagonal row in one game) to investigate how often this strategy is utilized. Results in Figure 6b illustrate that the frequency our agents use this strategy generally increases with training in the long term, indicating it is an acquired strategy rather than a circumstance.

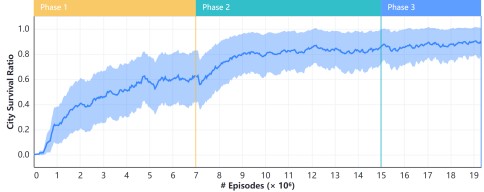 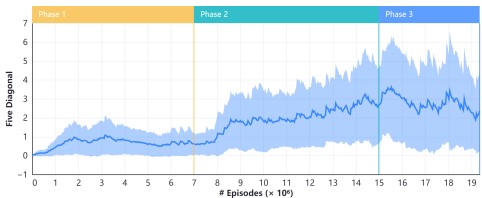

(a) **City survival ratio**: final number divided by the most number of *CityTiles* in one episode.

(b) **Five-diagonal**: how many times five or more *Workers* stand in a diagonal row in one episode.

Figure 6: **Quantitative analysis of region coordination using city survival ratio and five-diagonal.** Both metrics are evaluated using self-play. a) The city survival ratio increases as training continues, indicating that the regional construction planning effectively helps *CityTiles* live long and prosper. b) The frequency of the five-diagonal shape increases during the course of training, which demonstrates the gradual acquisition of this strategy.

## 5.3 GLOBAL STRATEGIES

As in micro-management scenarios of SMAC (Samvelyan et al., 2019), regional coordination of dozens of agents is often found in multi-agent cooperation. However, our agents go far beyond that, achieving much larger-scale coordination between hundreds of agents. We provide interpretation and analysis of several global strategies as follows:

**Sustainable Development.** A key component in Lux is the balance of city development and resource consumption. In the early stages, the rapidly growing cities often face severe fuel shortages. Gradually, our policy learns to develop cities at a sustainable speed in tune with fuel production depending on the resource distribution and the opponent's behavior. Another phenomenon we have observed is the retention of trees. As trees are the only renewable resource in Lux, forest protection is significant in securing long-term fuel supplies. Our agents intentionally preserve trees from excessive deforestation and build *CityTiles* near the woods in defense of the enemy's aggression. Another metric, total wood collect (the total collected woods divided by originally spawned woods) is used to measure how this forest protection strategy influences our fuel supplies. Results in Figure 7a show how these protection strategies significantly improve the utilization efficiency of wood, resulting in our agent collecting more than $500\%$ of the original wood on the map at times.

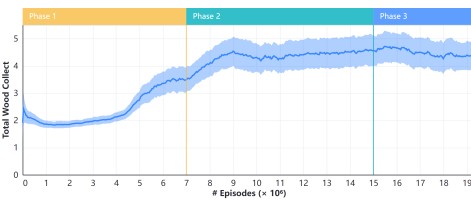 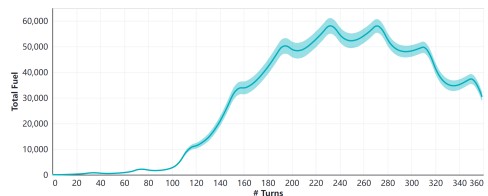

(a) **Total wood collect**: total wood collected divided by originally spawn woods.

(b) **Total fuel**: total fuel storage in one episode using our latest model in a fixed map.

Figure 7: **Quantitative analysis of global strategies using total wood collect and total fuel.** The metrics are evaluated using self-play. a) The utilization efficiency of wood increases as our policy grows the sense of forest protection. b) In one episode, our fuel storage accumulates until turn 240. After that, it tries to build more *CityTiles* for the win.

**All In For The Win.** Another surprising strategy is that when the episode is about to end, our policy will rapidly harvest all the protected trees and try to build as many *CityTiles* as possible for the win as shown in Figure 7b. Furthermore, we observe that sometimes cities retain very little fuel at the end of an episode, evidence that almost all the resources have been fully utilized, as any resources left at the end would be a waste. Efficiently using all remaining resources before the end is very challenging because it needs the overall calculation of total fuel consumption by all *Workers* and *CityTiles*, in addition to precise control of every agent. We think this strategy perfectly demonstrates the emergence of collective intelligence through the combination of long-term economic decisions and massive-agent mobilization.

# 6 EXPERIMENTS

In this section, we perform ablation studies to reflect on our policy implementation and general reinforcement learning algorithms under massive-agent settings: 1) we investigate the necessity of curriculum learning phases by training with different procedures. Results demonstrate that our curriculum design can help tackle the hard exploration problem caused by sparse rewards in the early stages of training and encourage the emergence of complex strategies beyond human design. 2) we further demonstrate the generalization ability of our model across different map sizes. When evaluating on maps of size 32, the policy trained on size 12 still retains some basic strategies. After a fine-tuning phase of only $1.8$ million episodes, the transferred policy achieves a $90\%$ win rate against TB on maps of size 32. The results indicate our model can learn generalizable representations suitable for the environment through learned spatial structures via convolutional layers. 3) we compare our centralized policy against a standard decentralized solution with carefully-designed team-based rewards. The centralized policy achieves a $98\%$ win rate. **See implementation details in Appendix A.2. The decentralized policy implementation and experiment are in Appendix A.3.**

## 6.1 DESIGN OF CURRICULUM LEARNING PHASES

We perform experiments to investigate the necessity of our curriculum learning phases. Five different procedures are applied: a) Only Phase 1; b) Phase 1 and 2 without Phase 3; c) Phase 1 and 3, without Phase 2; d) Phase 1, 2, and 3 (the original procedure); e) Only Phase 3.

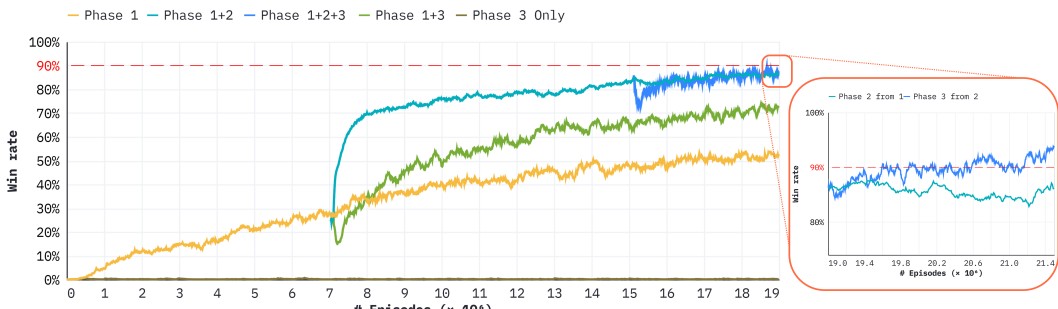

Figure 8: **The win rate curves from different training phases.** All win rates are evaluated against TB on maps of size 12. Training with direct sparse rewards results in a $0\%$ win rate, while training only with Phase 1 dense rewards converges at around $50\%$. Compared to Phase 1+2, Phase 1+3 improves slower and results in a lower win rate of around $70\%$. Phase 1+2 achieves an $85\%$ win-rate, and Phase 1+2+3 further boosts it to above $90\%$.

**Phase 1** utilizes dense rewards which are fundamental for helping the centralized policy acquire atomic skills for individual agents. As shown in Figure 8, our policy can hardly learn any basic skills when directly training with sparse rewards, resulting in a win rate of around zero due to the hard exploration problem.

**Phase 2** utilizes a scaled sparse reward which plays two roles in the whole learning procedure, accelerating learning and improving performance. First, continuous learning with dense rewards converges at a $50\%$ win rate, but the win rate rapidly rises to $70\%$ with Phase 2. On the other hand, without Phase 2, switching from Phase 1 to Phase 3 is more challenging with a lower performance even after training for a longer period. This shows that the scaled sparse reward can work as a proper transition between dense rewards and a win-or-loss sparse reward (applied in Phase 3) as it explicitly tells the policy that owning more cities is the key to winning.

**Phase 3** utilizes a sparse win-loss reward which further boosts the final performance to above $90\%$. As the Phase 2 training converges to a win rate of $85\%$ without Phase 3, the win-loss sparse reward pushes our policy to go further and explore, resulting in an overall $90\%$ win rate.

## 6.2 GENERALIZATION OF REPRESENTATIONS FOR REINFORCEMENT LEARNING

We provide clear evidence that the learned representations from the convolutional neural network and reinforcement learning algorithms can be generalized to different map sizes. First, we directly transfer the policy net trained on maps of size 12 to size 32. As shown in Figure 9, basic skills are retained on larger maps such as *Workers* collecting and fueling cities, even showcasing some structured city construction planning to surround and protect wood resources. Secondly, after an additional fine-tuning phase of around 1 million episodes, the policy quickly adapts to larger maps and achieves an overall $90\%$ win rate against TB, while training from scratch uses 1.6 million episodes for a $20\%$ win rate as shown in Figure 10.

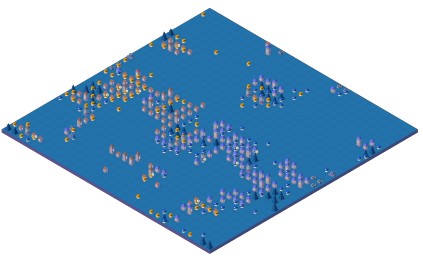

Figure 9: **Illustration of the generalization ability in a self-play episode.** When the policy trained on maps of size 12 is directly transferred to 32, some strategies are retained such as *Workers* building and fueling cities with plans.

Figure 10: **The win-rate curves on** $32 \times 32$ **maps of training from scratch and transfer.** Both are evaluated against TB. After a fine-tuning phase of 1 million steps, the transferred policy achieves a $90\%$ win rate, while training from scratch uses 1.6 million episodes for a $20\%$ win rate.

The results demonstrate the generalization ability of our model, which provides insights into speeding up policy training on large maps: we can pre-train our policy on small maps and then transfer it to large maps with a fine-tuning phase. This procedure significantly reduces the training time because smaller maps are faster for environment simulation and network update. For example, the Lux environment simulation on CPU is $2.5\times$ slower on maps of size 32 than size 12, and the policy network update on GPU is $5\times$ slower on maps of size 32. As training on maps of size 12 is both time-saving and computationally efficient, our "Pre-train and Fine-tune" scheme achieves a higher win rate with fewer training hours.

## 7 DISCUSSION AND FUTURE WORK

We have demonstrated that collective intelligence can emerge from massive-agent cooperation and competition. As proof of concept, we propose Lux, an environment hosting hundreds of agents and incorporating RTS game dynamics. Through standard reinforcement learning algorithms and pixel-to-pixel centralized modeling, we observe several stages of agents' strategy evolution. Our agents exhibit ambitious group strategies based on accurate individual control of massive agents without explicit coordination mechanisms, signifying the emergence of collective intelligence.

We hope our work with Lux will be a stepping stone toward artificial collective intelligence. In Lux, we observe the number of agents can reach up to 2000 in a single timestep, but this still pales in comparison to the millions or even billions of organisms cooperating and competing in nature. The Lux environment can be easily extended to host more agents as the experiments in Appendix A.4, but simulation and inference become extremely slow reaching the million-agent level. Going forward, the environment design and engineering as well as the training algorithm need a lot of modifications to adapt to such a scale. We also acknowledge that the RTS game dynamics in Lux may not directly coincide with real-world problems. However, with Lux as a blueprint, economic rules and dynamics like Zheng et al. (2021) can be incorporated, which may provide some reference for economic decisions and policies in the real world.

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

# A  APPENDIX

## A.1  DETAILED RULES OF LUX

For ease of understanding, the environment rules including unit types and action spaces are simplified in the main text. In this part, we provide a detailed description of the environment design and rules. The version of Lux we use is compatible with the version on Kaggle Lux AI S1 competition[2]. The rules can also be found at https://www.lux-ai.org/specs-2021 and the following text is a reformatted and slightly modified version of the original rules.

**Background.** The night is dark and full of terrors. Two teams must fight off the darkness, collect resources, and advance through the ages. Daytime finds a desperate rush to gather and build the resources to carry you through the impending night. Plan and expand carefully – any city that fails to produce enough light will be consumed by darkness.

**Environment.** In the Lux AI Challenge Season 1, two competing teams control a team of Units and *CityTiles* that collect resources to fuel their Cities, with the main objective to own as many *CityTiles* as possible at the end of the turn-based game. Both teams have complete information about the entire game state and use that information to optimize resource collection, compete for scarce resources against the opponent, and build cities to gain points. Each competitor must program their policy in their language of choice. Each turn, your agent gets 3 seconds to submit their actions, excess time is not saved across turns. In each game, you are given a pool of 60 seconds that is tapped into each time you go over a turn's 3-second limit. Upon using up all 60 seconds and going over the 3-second limit, your agent freezes and can no longer submit additional actions.

**The Map.** The world of Lux is represented as a 2D grid. Coordinates increase east (right) and south (down). The map is always a square and can be 12, 16, 24, or 32 tiles long. The $(0,0)$ coordinate is at the top left.

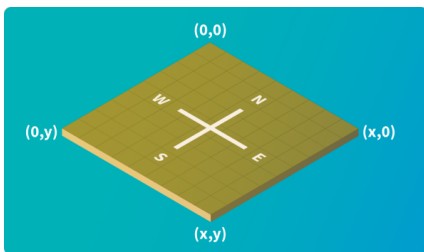

Figure 11: The specification of the map in Lux.

The map has various features including Resources (Wood, Coal, Uranium), Units (Workers, Carts), *CityTiles*, and Roads. In order to prevent maps from favoring one player over another, it is guaranteed that maps are always symmetric by vertical or horizontal reflection. Each player will start with a single *CityTile* and a single *Worker* on that *CityTile*.

**Resources.** There are 3 kinds of resources: Wood, Coal, and Uranium (in order of increasing fuel efficiency). These resources are collected by *Workers*, then dropped off once a *Worker* moves on top of a *CityTile* to then be converted into fuel for the city. Some resources require research points before they are possible to collect. Wood in particular can regrow. Each turn, every wood tile's wood amount increases by $2.5\%$ of its current wood amount rounded up. Wood tiles that have been depleted will not regrow. Only wood tiles with less than $500$ wood will regrow.

---

[2]https://www.kaggle.com/c/lux-ai-2021/

Table 1: The specifications of resource collection and convert.

| Resource Type | Research Points Pre-requisite | Fuel Value per Unit | Units Collected per Turn |
|---|---|---|---|
| Wood | 0 | 1 | 20 |
| Coal | 50 | 10 | 5 |
| Uranium | 200 | 40 | 2 |

**Collection Mechanics.** At the end of each turn, *Workers* automatically receive resources from all adjacent (North, East, South, West, or Center) resource tiles they can collect resources from according to the current symmetric formula:

Iterating over uranium, coal, then wood resources:

- Each unit makes resource collection requests to collect an even number of resources from each adjacent tile of the current iterated resource such that the collected amount takes the unit's cargo above capacity. E.g. *Worker* with 60 wood adjacent to 3 wood tiles asks for 14 from each, receives 40 wood, and wastes 2.

- All tiles of the current iterated resource then try to fulfill requests. If they can't, they make sure all unfulfilled requests get an equal amount, and the leftover is wasted. E.g. if 4 *Workers* are mining a tile of 25 wood, but one of them is only asking for 5 while the others are asking for 20 wood each, then first all *Workers* get 5 wood each, leaving 5 wood left over for 3 more *Workers* with space left. This can be evenly distributed by giving 1 wood each to the last 3 *Workers*, leaving 2 wood left that is then wasted.

Workers cannot mine while on *CityTiles*. Instead, if there is at least one *Worker* on a *CityTile*, that *CityTile* will automatically collect adjacent resources at the same rate as a *Worker* each turn and directly convert it all to fuel. The collection mechanic for a *CityTile* is the same as a *Worker* and you can treat a *CityTile* as an individual *Worker* collecting resources.

**Actions.** Units and *CityTiles* can perform actions each turn given certain conditions. In general, all actions are simultaneously applied and are validated against the state of the game at the start of a turn. The next few sections describe the Units and *CityTiles* in detail.

**CityTiles.** A *CityTile* is a building that takes up one tile of space. Adjacent *CityTiles* collectively form a City. Each *CityTile* can perform a single action provided the *CityTile* has a Cooldown $< 1$.

Actions:

- Build *Worker* - Build *Worker* unit on top of this *CityTile* (cannot build a *Worker* if the current number of owned *Workers* + carts equals the number of owned *CityTiles*).

- Build Cart - Build Carts unit on top of this *CityTile* (cannot build a cart if the current number of owned *Workers* + carts equals the number of owned *CityTiles*).

- Research - Increase your team's Research Points by 1.

**Units.** There are two unit types, *Workers*, and Carts. Every unit can perform a single action once they have a Cooldown $< 1$. All units can choose the move action and move in any of the 5 directions, North, East, South, West, or Center. Moreover, all units can carry raw resources gained from automatic mining or resource transfer. *Workers* are capped at 100 units of resources and Carts are capped at 2000 units of resources. Whenever a unit moves on top of a friendly *CityTile*, the City that *CityTile* forms converts all carried resources into fuel.

There can be at most one unit on tiles without a *CityTile*. Moreover, units cannot move on top of the opposing team's *CityTiles*. However, units can stack on top of each other on a friendly *CityTile*. If two units attempt to move to the same tile that is not a *CityTile*, this is considered a collision, and the move action is canceled.

**Workers.** Actions:

- Move - Move the unit in one of 5 directions, North, East, South, West, or Center.
- Pillage - Reduce the Road level of the tile the unit is on by $0.5$.

Table 2: The specifications of cooldown.

| Unit Type | Base Cooldown |
|-----------|---------------|
| CityTile  | 10            |
| Worker    | 2             |
| Cart      | 3             |

Table 3: The specifications of file burn, $n_{adj}$ denotes the number of adjacent friendly *CityTiles*.

| Unit Type | Fuel Burn in City | Fuel Burn Outside City |
|-----------|-------------------|------------------------|
| CityTile  | $23 - 5 \times n_{adj}$ | N/A |
| Worker    | 0                 | 10                     |
| Cart      | 0                 | 4                      |

- Transfer - Send any amount of a single resource-type from a unit's cargo to another (start-of-turn) adjacent Unit, up to the latter's cargo capacity. Excess is returned to the original unit.

- Build *CityTile* - Build a *CityTile* right under this *Worker*, provided the *Worker* has 100 total resources of any type in their cargo (full cargo), and the tile is empty. If the building is successful, all carried resources are consumed, and a new *CityTile* is built with 0 starting resources.

**Carts.** Actions:

- Move - Move the unit in one of 5 directions, North, East, South, West, Center.

- Transfer - Send any amount of a single resource-type from a unit's cargo to another (start-of-turn) adjacent Unit, up to the latter's cargo capacity. Excess is returned to the original unit.

**Cooldown.** *CityTiles*, *Workers*, and Carts all have a cooldown mechanic after each action. Units and *CityTiles* can only act when they have Cooldown $< 1$. At the end of each turn, after Road has been built and pillaged, each unit's Cooldown decreases by 1 and decreases by the level of the Road the unit is on at the end of the turn. *CityTiles* are not affected by road levels, and cooldown always decreases by 1. The minimum Cooldown is 0. After an action is performed, the unit's Cooldown will increase by a Base Cooldown, as specified in Table 2.

**Roads.** As Carts travel across the map, they start to create roads that allow all Units to move faster. At the end of each turn, Cart will upgrade the road level of the tile it ends on by $0.75$. The higher the road level, the faster Units can move and perform actions. All tiles start with a road level of 0 and are capped at 6. Moreover, *CityTiles* automatically have a max road level of 6. *Workers* can also destroy roads via the pillage action which reduces road levels by $0.5$ each time. If a City is consumed by darkness, the road level of all tiles in the City's *CityTiles* will go back to 0.

**Day/Night Cycle.** The Day/Night cycle consists of a 40-turn cycle, the first 30 turns being day turns, the last 10 being night turns. There are 360 turns in a match, forming 9 cycles. During the night, Units and Cities need to produce light to survive. Each turn of the night, each Unit and *CityTile* will consume an amount of fuel, see Table 3 for rates. Units in particular will use their carried resources to produce light whereas *CityTiles* will use their fuel to produce light. *Workers* and Carts will only need to consume resources if they are not on a CityTile. When outside the City, *Workers* and Carts must consume whole units of resources to satisfy their night needs, e.g. if a *Worker* carries 1 wood and 5 uranium on them, they will consume a full wood for 1 fuel, then a full unit of uranium to fulfill the last 3 fuel requirements, wasting 37 fuel. Units will always consume the least efficient resources first.

Lastly, at night, Units gain $2\times$ more Base Cooldown. Should any Unit during the night run out of fuel, they will be removed from the game and disappear into the night forever. Should a City run out of fuel, however, the entire City with all of the *CityTiles* it owns will fall into darkness and be removed from the game.

**Game Resolution order.** To help avoid confusion over smaller details of how each turn is resolved, we provide the game resolution order here and how actions are applied. Actions in the game are first all validated against the current game state to see if they are valid. Then the actions, along with game events, are resolved in the following order and simultaneously within each step:

1. *CityTile* actions along with increased cooldown.
2. Unit actions along with increased cooldown.
3. Roads are created.
4. Resource collection.
5. Resource drops on *CityTiles*.
6. If night time, make Units consume resources and *CityTiles* consume fuel.
7. Regrow wood tiles that are not depleted to $0$.
8. Cooldowns are handled/computed for each unit and *CityTile*.

The only exception to the validation criteria is that units may move smoothly between spaces, meaning if two units are adjacent, they can swap places in one turn. Otherwise, actions such as one unit building a *CityTile*, then another unit moving on top of the new *CityTile*, are not allowed as the current state does not have this newly built city and units cannot move on top of other units outside of *CityTiles*.

**Win Conditions.** After $360$ turns the winner is whichever team has the most *CityTiles* on the map. If that is a tie, then whichever team has the most units owned on the board wins. If still a tie, the game is marked as a tie. A game may end early if a team no longer has any more Units or *CityTiles*. Then the other team wins.

### A.2   ADDITIONAL IMPLEMENTATION DETAILS

Detailed information on our policy implementation is illustrated in this section, including feature engineering, network design, and reinforcement learning algorithm implementation.

**PPO implementation.** Standard PPO loss (Schulman et al., 2017) is used as the policy loss to optimize the policy net and to estimate the advantage, we apply GAE (Schulman et al., 2016) with the trajectory length as 32. For Phase 1 training with dense rewards, we set the GAE parameter $\lambda = 0.95$, the discount factor $\gamma = 0.99$. For Phase 2 and 3 training with sparse rewards, we set $\lambda = 1, \gamma = 1$. We apply PPO2 with a clip operation and set the clipping ratio $\epsilon = 0.1$. Mean squared loss is used to optimize the critic's value head and the weight of the value loss is set as $0.5$. The entropy loss is also computed to encourage exploration with a coefficient of $0.1$. We use Adam (Kingma & Ba, 2014) as the optimizer with the learning rate $1e - 4$. For computational resources, we use an NVIDIA-V100 GPU and 600 CPU cores.

**Input.** Input features fed into our policy network consists of two parts: 1) the vector containing global information such as timestep and total fuel. Features in the global information vector and their corresponding data specifications are listed in Table 4. 2) the image input containing the map information and the locations of our own and enemy's *Workers* and *CityTiles* are listed in Table 5.

**Dense Reward Design.** The detailed dense reward design is listed in Table 6. Four types of rewards are given for specific behaviors of *CityTiles* and *Workers*. The total reward is the sum of four sub-rewards.

**Data Preprocessing.** The whole data preprocessing procedure is illustrated in Figure 12. The global information vector is split into two parts, the one-hot features of dimension $51$ and the other features of dimension $18$. The one-hot features are fed into a linear layer with an output dimension of $9$ for embedding. For unifying input into image shapes, the embedding vector of length $9$ and the other global information vector of length $18$ are expanded as the image sizes in separate channels, where every pixel in each channel is of the same value. After that, these expanded image features ($9 \times H \times W$ and $18 \times H \times W$, $H$ and $W$ are the map height and width) along with the original image features ($37 \times H \times W$) are passed through separate convolutional neural networks with a kernel size of 1. Then three parts of input images are concatenated together in the channel dimension. Through

Table 4: Input features Part 1: global information vector.

| Feature Description | Type | Range | Normalization Coefficient |
|---|---|---|---|
| Current Cycle | One-Hot | [0,8] | N/A |
| Current Turn In This Cycle | One-Hot | [0,39] | N/A |
| If At Night | One-Hot | [0,1] | N/A |
| Own *CityTile* Numbers | Int | [0,1024] | 100 |
| Enemy *CityTile* Numbers | Int | [0,1024] | 100 |
| Own Unit Numbers | Int | [0,1024] | 100 |
| Enemy Unit Numbers | Int | [0,1024] | 100 |
| Own Research Points | Int | [0,200] | 200 |
| Enemy Research Points | Int | [0,200] | 200 |
| Own Total Fuel | Int | $[0,10^6]$ | 2300 |
| Enemy Total Fuel | Int | $[0,10^6]$ | 2300 |
| Own Average Fuel Per *CityTile* | Float | $[0,10^4]$ | 230 |
| Enemy Average Fuel Per *CityTile* | Float | $[0,10^4]$ | 230 |
| Own Total Fuel Cost | Int | $[0,10^5]$ | 230 |
| Enemy Total Fuel Cost | Int | $[0,10^5]$ | 230 |
| Own Average Fuel Cost Per *CityTile* | Float | [0,23] | 23 |
| Enemy Average Fuel Cost Per *CityTile* | Float | [0,23] | 23 |
| If Own Team Can Collect Coal | Bool | [0,1] | N/A |
| If Enemy Team Can Collect Coal | Bool | [0,1] | N/A |
| If Own Team Can Collect Uranium | Bool | [0,1] | N/A |
| If Enemy Team Can Collect Uranium | Bool | [0,1] | N/A |

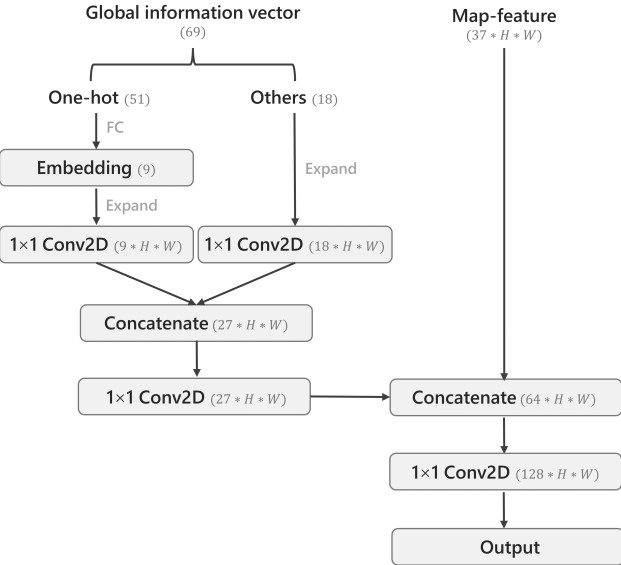

Figure 12: **The Data Preprocessing Procedure**. Global information is split into two parts, one-hot and others. One-hot vector is first embedded through a fully-connected layer and then expands as the map size. After passing the $1 \times 1$ Conv2D, one-hot and other features are concatenated in channels, going through another $1 \times 1$ Conv2D. The global features and map features are concatenated with another $1 \times 1$ Conv2D with input channels as 64 and output channels as 128.

another convolutional layer with a kernel size of 1, input channels as 64, and output channels as 128, the tensors as the ResNet backbone input is in shape $128 \times H \times W$.

Table 5: Input features Part 2: image features.

| Feature Description | Type | Normalization Coefficient |
|---|---|---|
| If No *Worker* | Bool | N/A |
| If Own *Worker* | Bool | N/A |
| If Enemy *Worker* | Bool | N/A |
| If No Cart | Bool | N/A |
| If Own Cart | Bool | N/A |
| If Enemy Cart | Bool | N/A |
| If No *CityTile* | Bool | N/A |
| If Own *CityTile* | Bool | N/A |
| If Enemy *CityTile* | Bool | N/A |
| Road Level | Float | 6 |
| Worker Cooldown | Float | 10 |
| If *Worker* Can Act | Bool | N/A |
| Cart Cooldown | Float | 10 |
| If Cart Can Act | Bool | N/A |
| CityTile Cooldown | Float | 10 |
| If *CityTile* Can Act | Bool | N/A |
| If It is Resource | Bool | N/A |
| Wood Amount | Int | 100 |
| If Wood Can Regrow | Bool | N/A |
| Coal Amount | Int | 100 |
| Uranium Amount | Int | 100 |
| Worker Wood Carry Amount | Int | 100 |
| Worker Coal Carry Amount | Int | 100 |
| Worker Uranium Carry Amount | Int | 100 |
| If *Worker* Reaches Carry Limit | Bool | N/A |
| Cart Wood Carry Amount | Int | 100 |
| Cart Coal Carry Amount | Int | 100 |
| Cart Uranium Carry Amount | Int | 100 |
| CityTile Fuel Cost | Int | 100 |
| CityTile Average Fuel | Float | 230 |
| If *CityTile* Can Survive Tonight | Bool | N/A |
| Fuel *CityTile* Needed to Survive | Int | 230 |
| If *Worker* is at *CityTile* | Bool | N/A |
| If Cart is at *CityTile* | Bool | N/A |
| X Relative Distance to Center | Float | N/A |
| Y Relative Distance to Center | Float | N/A |

Table 6: Design details of Phase 1: dense rewards.

| Units | Behaviors | Weights |
|---|---|---|
| CityTiles | Research Points Increases | 0.01 |
| | Units Built | 0.5 |
| Workers | Fuel Increases | 0.0001 |
| | *CityTiles* Built | 1 |

**ResNet Backbone.** The ResNet backbone consists of 8 Residual blocks. Each Residual block comprises two convolutional layers and a Squeeze-and-Excitation(SE) layer. The detailed structure of the Residual Block is shown in Figure 13.

**Output.** After the ResNet backbone, we use multiple heads for the output actions and value. The learned representation from the ResNet backbone is first passed through a Spectral Normalization layer. For the action head, we use three separate heads for the *Workers*, *Carts* and *CityTiles*. Each head is a convolutional layer with kernel size as 1 and output channels as the corresponding action

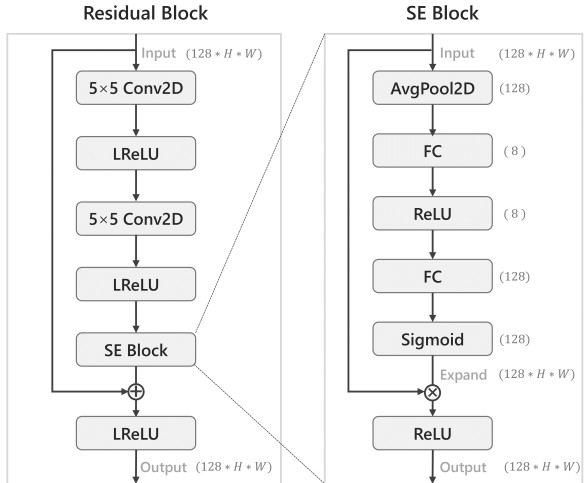

Figure 13: **Residual Block Design and Squeeze-and-Excitation Layer.** Each Residual block consists two convolutional layers (kernel size $= 5$, padding $= 2$, stride $= 1$) and LeakyReLU as the activation function. Squeeze-and-Excitation(SE) layer consists of a 2D Average Pooling, and two fully-connected layers.

dimensions (19 for *Worker*, 17 for *Cart* and 4 for *CityTile*). For the critic's head, we use an Average Pooling to transform the representation of size $128 \times H \times W$ to a vector of length 128. Then we use a fully-connective layer to get a single value for the critic's estimation.

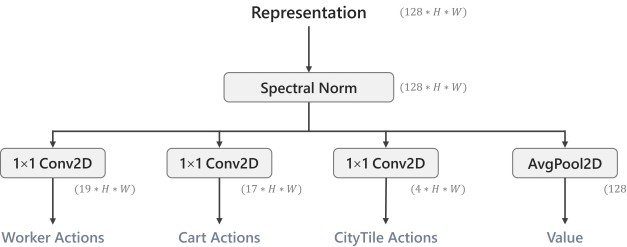

Figure 14: **Output actions and value.** First through a spectral normalization layer, three action heads, and a value head are appended. $1 \times 1$ Conv2D is used for output actions of *Worker*, Cart, and *CityTile*. AvgPool2D and a fully-connected layer are used for value estimation.

**Valid Action Mask.** Valid action mask is a common technique in reinforcement learning to eliminate unnecessary explorations and accelerate the learning process. We calculate the valid action mask based on the following rules:

- *Workers*: All actions are invalid when cooldown $> 1$. When cooldown $< 1$, moving to enemy *CityTiles* or tiles with other *Workers* on it is invalid; moving to friendly *CityTile* is always valid; and building a *CityTile* is valid only when its resource achieves 100. Doing nothing is always valid.

- *CityTiles*: All actions are invalid when cooldown $> 1$. When cooldown $< 1$, building a *Worker* is valid when the number of workers are less than the number of *CityTiles*; research is valid when the team's research point $< 200$. Doing nothing is always valid.

## A.3 DECENTRALIZED POLICY IMPLEMENTATION AND ABLATION STUDIES

In this section, detailed information on our decentralized policy implementation is described, including input features, network design, and rule-based agents.

Table 7: Decentralized policy input: global information.

| Feature Description | Type | Range |
|---|---|---|
| Number of Agents Observed | Int | 320 |
| Global *CityTile* Number | One-hot | [0,320] |
| Global Unit Number | One-hot | [0,320] |
| Current Cycle | One-hot | [0,9] |
| Current Turn in this cycle | One-hot | [0,39] |
| If At Night | Bool | N/A |
| Own Research Point | Int | [0,200] |
| If Own Team Can Collect Coal | Bool | N/A |
| If Own Team Can Collect Uranium | Bool | N/A |

Table 8: Decentralized policy input: self information.

| Feature Description | Type | Range |
|---|---|---|
| Location X | Int | [0,31] |
| Location Y | Int | [0,31] |
| Location X | One-Hot | [0,31] |
| Location Y | One-Hot | [0,31] |
| Type | Bool | N/A |
| If At City | Bool | N/A |
| Alive | Bool | [N/A |
| Cooldown | One-Hot | [0,9] |
| If At Night | Bool | N/A |
| Wood Carry Amount | Int | [0,100] |
| Coal Carry Amount | Int | [0,100] |
| Uranium Carry Amount | Int | [0,100] |
| Wood Carry Amount | One-Hot | [0,100] |
| Coal Carry Amount | One-Hot | [0,100] |
| Uranium Carry Amount | One-Hot | [0,100] |
| Fuel | Int | [0,4000] |
| Fuel | One-Hot | [0,4000] |

**Input.** The input of our decentralized policy can be divided into four parts: global information (listed in Table 7), self information (listed in Table 8), other agents (team and enemy) information (listed in Table 9) and map information (listed in Table 10).

**Reward Shaping.** Each agent receives three types of rewards: 1) Its own reward to encourage certain behaviors such as survival, building cities, collecting resources, and fueling cities. 2) *CityTile* reward. Though the *CityTile* is a rule-based reward, this reward is used to guide the *Workers'* behavior to support the *CityTiles*. 3) Team reward. It consists of a final win reward and average reward of the team to encourage cooperation among agents.

**Network Design.** The input is split into seven parts, i.e., global features, self features, friend *Worker* features, friend *CityTile* features, enemy *Worker* features, enemy *CityTile* features, and image features. For the former six vector features, we use six different two-layer fully-connected networks for feature extraction. And for those features involving multiple units, we perform max pooling along units. For the image features, we use three convolutional layers and flatten the learned representations. Then those representations are concatenated together and passed through two fully-connected layers for the actions and values.

**Rule-based Agents.** For simplicity, we only use *Worker* as reinforcement learning agents and *CityTiles* as rule-based agents. The decision rules of *CityTiles* are simple and intuitive: 1) Build a *Worker* if a *CityTile* can. 2) If it cannot build a *Worker* and the team's research point $< 200$, research. 3) Do nothing otherwise.

Table 9: Decentralized policy input: other agents' information.

| Other Agents | Feature Description | Type | Range |
|---|---|---|---|
| Own *Worker* × 160 | Is Friend | Bool | N/A |
| | Location X | Int | [0,31] |
| | Location Y | Int | [0,31] |
| | Distance | Int | [0,62] |
| | If At City | Bool | N/A |
| | Cooldown | One-Hot | [0,3] |
| | Wood Carry Amount | Int | [0,100] |
| | Coal Carry Amount | Int | [0,100] |
| | Uranium Carry Amount | Int | [0,100] |
| Own *CityTile* × 160 | Is Friend | Bool | N/A |
| | Location X | Int | [0,31] |
| | Location Y | Int | [0,31] |
| | Distance | Int | [0,62] |
| | Cooldown | One-Hot | [0,9] |
| | Average Fuel Per *CityTile* | Float | [0,2300] |
| | Fuel Cost Per Night | Int | [0,23] |
| | If Can Survive Tonight | Bool | N/A |
| | Fuel Needed to Survive Tonight | Int | [0,230] |
| Enemy *Worker* × 160 | Is Friend | Bool | N/A |
| | Location X | Int | [0,31] |
| | Location Y | Int | [0,31] |
| | Distance | Int | [0,62] |
| | If At City | Bool | N/A |
| | Cooldown | One-Hot | [0,3] |
| | Wood Carry Amount | Int | [0,100] |
| | Coal Carry Amount | Int | [0,100] |
| | Uranium Carry Amount | Int | [0,100] |
| Enemy *CityTile* × 160 | Is Friend | Bool | N/A |
| | Location X | Int | [0,31] |
| | Location Y | Int | [0,31] |
| | Distance | Int | [0,62] |
| | Cooldown | One-Hot | [0,9] |
| | Average Fuel Per *CityTile* | Float | [0,2300] |
| | Fuel Cost Per Night | Int | [0,23] |
| | If Can Survive Tonight | Bool | N/A |
| | Fuel Needed to Survive Tonight | Int | [0,230] |

**Compared with Decentralized Control.** To illustrate our pixel-to-pixel centralized control solution, we perform a comparative experiment with the decentralized control solution. Competing with the decentralized control solution, the centralized control solution achieves 98% win rate computing by 100 runs. As shown in Figure 16, the decentralized policy can acquire basic skills such as collecting and building *CityTiles*. Encouraged by the team-based reward, decentralized agents even acquired a basic level of regional cooperation. However, since the cooperation is induced by pre-engineered rewards, it can only be applied to special scenarios and cannot be extended to other complex situations. For example, in a local map with woods, the decentralized agents are at an advantage initially, but due to their cooperation lacking adaptivity, our agents gradually build cities surrounding them and limit their development to gain the advantage. As a result, at Turn 120, the centralized policy has taken control of every resource on the map. Moreover, more group strategies emerged from the evolution of the centralized policy, for instance, being aggressive in sending some *Workers* to occupy and protect the key resources from its opponent.

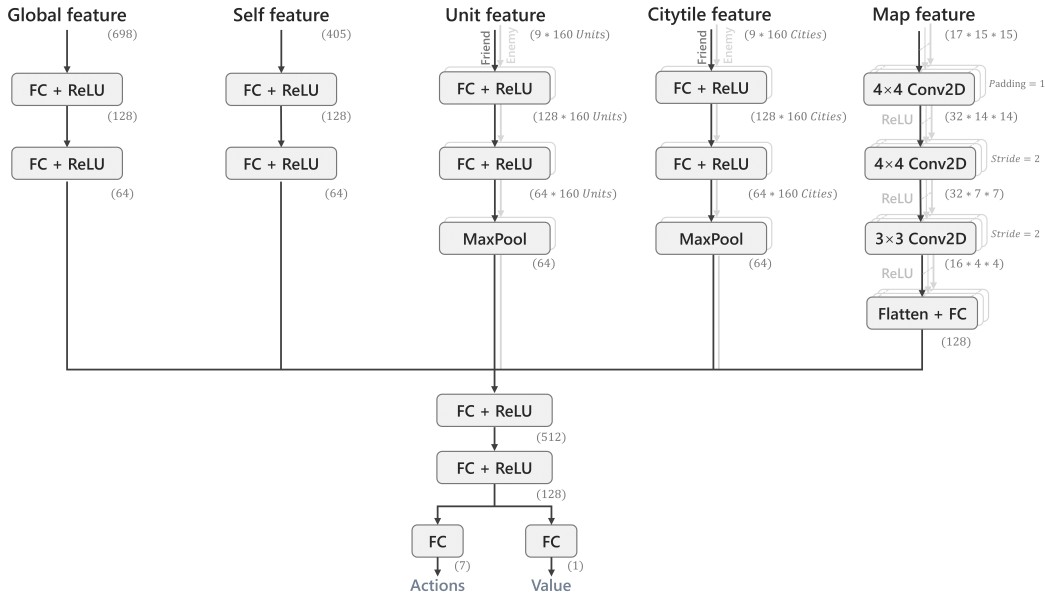

Figure 15: **Decentralized policy net architecture.** For the global and self features, we use two fully-connected layers for extraction. For unit features and *CityTile* features, we use two linear layers and then apply MaxPooling along the units. Three convolutional layers with a flattened and linear layer are used for map features. Then the outputs are concatenated together, using two fully-connected layers for the output actions and value.

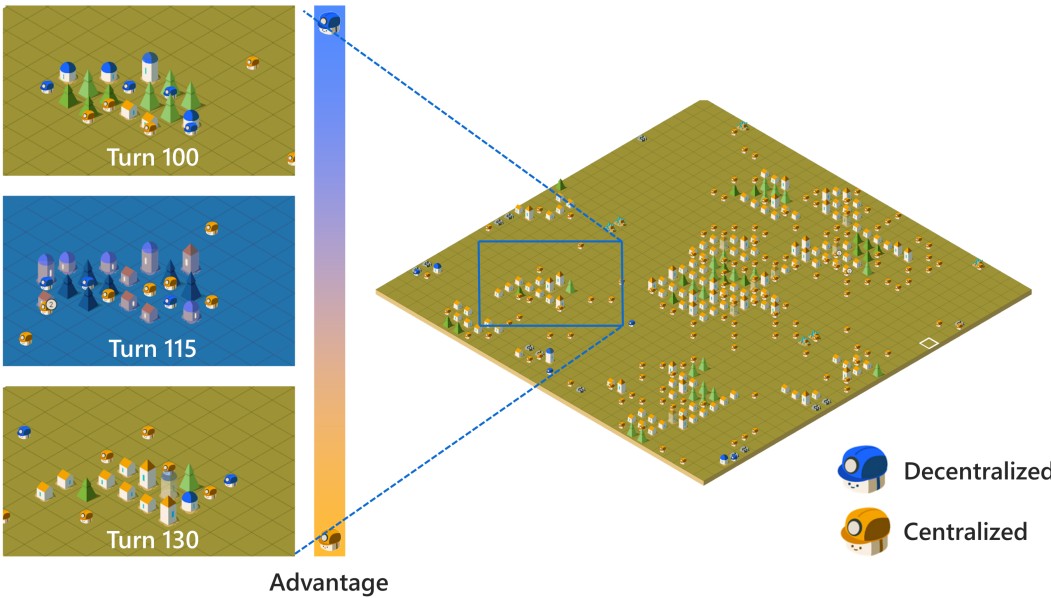

Figure 16: **One episode between the decentralized and centralized policy.** Yellow is the decentralized policy, and Blue is the centralized policy. In a local battle, Blue is at an advantage at first, but with better coordination, Yellow turns things around within just 30 turns.

Table 10: Decentralized policy input: map information.

| Map Information | Feature Description | Type | Range |
|---|---|---|---|
| Resource Map | Is Wood Here | Bool | N/A |
| | Wood Reserves | Int | [0,1000] |
| | Is Coal Here | Bool | N/A |
| | Coal Reserves | Int | [0,1000] |
| | Is Uranium Here | Bool | N/A |
| | Uranium Reserves | Int | [0,1000] |
| Worker Map | Is Friend *Worker* | Bool | N/A |
| | Is Enemy *Worker* | Bool | N/A |
| | *Worker* Cooldown | Int | [0,3] |
| | *Worker* Wood Carry Amount | Int | [0,100] |
| | *Worker* Coal Carry Amount | Int | [0,100] |
| | *Worker* Uranium Carry Amount | Int | [0,100] |
| City Map | Is Friend *CityTile* | Bool | N/A |
| | Is Enemy *CityTile* | Bool | N/A |
| | *CityTile* Cooldown | Int | [0,9] |
| | Average Fuel Per *CityTile* | Float | [0,2300] |
| | Fuel Cost Per Night | Int | [0,23] |
| | If Can Survive Tonight | Bool | N/A |
| | Fuel Needed to Survive Tonight | Int | [0,230] |
| | Road Level | Bool | N/A |

Table 11: Decentralized policy reward design.

| Reward Type | Feature Description | Weights |
|---|---|---|
| Worker Reward | *Worker* Death Penalty | $-1$ |
| | *Worker* Survive One Night Turn | 0.05 |
| | *Worker* Survive Ten Night Turns | 0.5 |
| | *Worker* Build a *CityTile* | 1 per *CityTile* |
| | Built City Fuel Saving | $0.05 \times (23 - \text{Fuel Cost})$ |
| | *Worker* Fuel Increase | 0.005 per fuel |
| | *Worker* Fuel Donation | 0.01 per fuel |
| CityTile Reward | *CityTile* Death penalty | $-1$ |
| | *CityTile* Survive One Night Turn | 0.05 |
| | *CityTile* Survive Ten Night Turns | 0.5 |
| | *CityTile* Research Point Increase | 0.02 per point |
| | Research Point reaches 50 | 1 |
| | Research Point reaches 200 | 4 |
| | *CityTile* Build a *Worker* | 1 |
| Team Reward | Final Win Reward | 100 |
| | Team Average Reward | $0.1\times$ average of friend reward |

## A.4 MORE GENERALIZATION STUDIES

In section 6.2, we demonstrate the generalization of our model by transferring the policy trained on maps of size 12 to size 32. More studies are conducted to further investigate the generalization ability of our proposed model on larger maps. Results show that even transferred to larger maps, our model still retains a surprising ability of massive-agent coordination.

We use the model trained on $32 \times 32$ maps as the base model and evaluate it on different map sizes without fine-tuning. On maps of sizes 48 and 64, our policy shows the fantastic mastery of massive-agent coordination as shown in Figure 17.

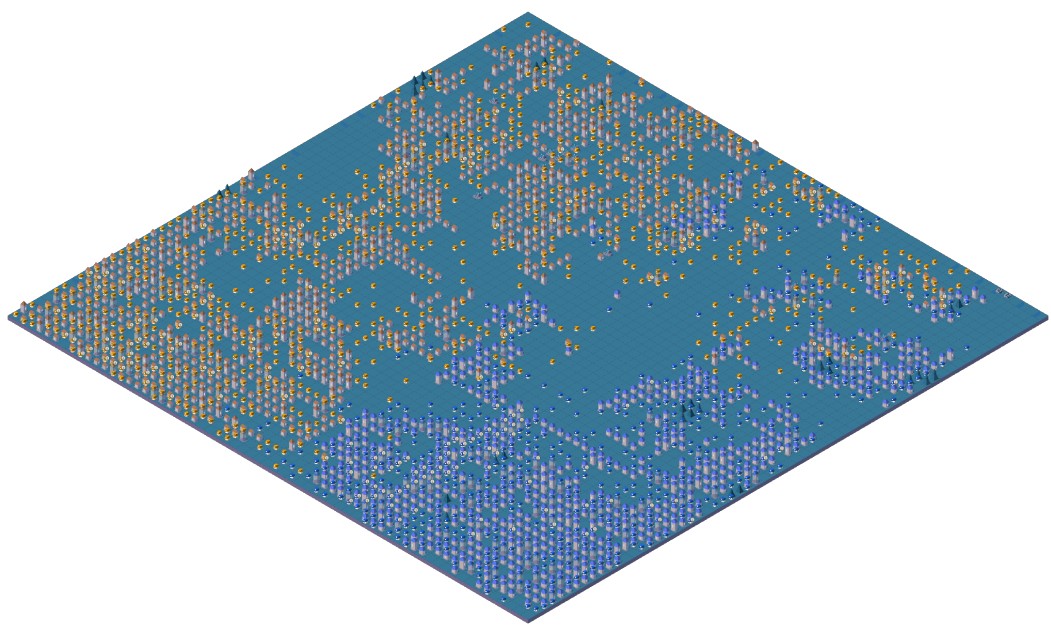

Figure 17: **Policy transfer on maps of size 64.** There are 1069 *CityTiles* and 1059 units for the orange team, and 779 *CityTiles* and 768 units for the blue team. In larger maps, our policy demonstrates the generalization ability of coordination between thousands of agents.

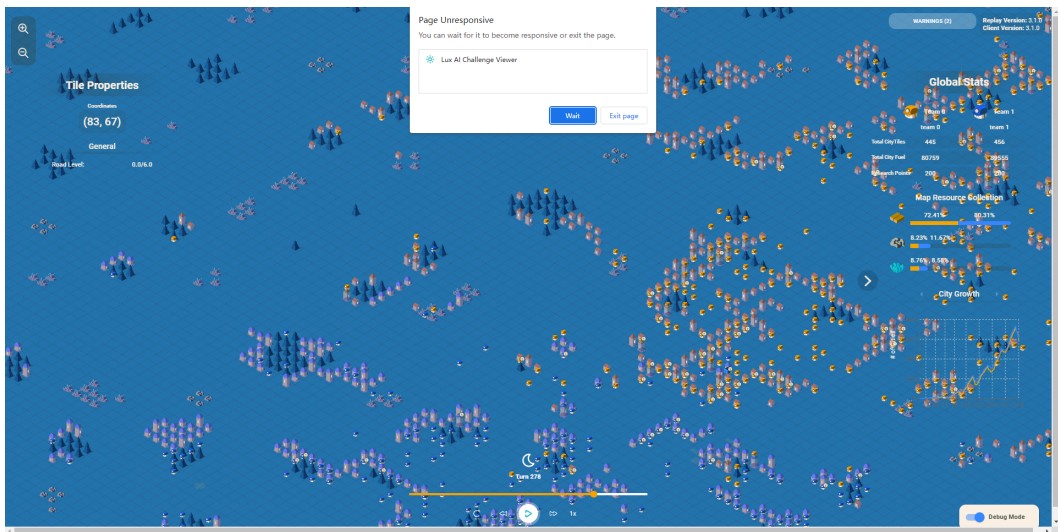

Figure 18: **Policy transfer on maps of size 128.** The large map and the cooldown mechanism limit the ability to build large cities fulfilling the map. However, our policy still exhibits skills and strategies they acquire on $32 \times 32$ maps. This large-scale setting eventually causes the web viewer unresponsive.

Furthermore, we make a bold attempt on the $128 \times 128$ maps. However, due to the cooldown mechanism, agents can hardly travel across the map within 360 turns, which makes it impossible to build large cities fulfilling the map like they do in $32 \times 32$ maps. Moreover, in a larger map, the environment simulation is much slower, which takes about 30 minutes for one episode. It indicates that although Lux has the scalability for millions of agents, the game core and the dynamics need a lot of modification to adapt to larger scales. Nevertheless, we find our agents still exhibit skills and strategies they acquire on $32 \times 32$ maps as in Figure 18, which demonstrates the potential of our method at a million-agent scale.

