# OpenReview forum: "Emergent collective intelligence from massive-agent cooperation and competition"
_ICLR.cc/2023/Conference — Submitted to ICLR 2023_

### Official Review · Reviewer_evMn · 2022-10-23

**Confidence:** 4
**Correctness:** 3
**Technical Novelty And Significance:** 2
**Empirical Novelty And Significance:** 2
**Recommendation:** 3

**Clarity, Quality, Novelty And Reproducibility:**



The paper is not well structured. Key parts are in the Appendix, including the definition of the notation used in the paper.

The reviewer believes that the novelty of this work is rather limited. The implementation of the environment itself is rather standard.

The emerging patterns in terms of strategy are interested, but they appear as a direct consequence of the reward structure defined by the authors. The reviewer believes that the contribution itself is not sufficient for a conference of the level of ICLR.

The study using curriculum learning is also rather standard for this type of environments. There are limited novel insights that are emerging in this analysis.

The authors should carefully proofread the paper. There are missing citations and wrong capitalization in the text. The presentation of the evaluation results in terms of description of the experimental settings can also be improved. The discussion of the experimental results is rather concise.


**Strength And Weaknesses:**

General Note

It appears that the paper is not anonymized since the authors say "To this end, we propose Lux (Doerschuk-Tiberi & Tao, 2021), a cooperative and competitive environment where hundreds of agents in two populations scramble for limited resources and fight off the darkness". It appears that the authors of the work are linked to Doerschuk-Tiberi & Tao.

Strengths

- The release of a platform for the research community is commendable.
- The emerging patterns are interesting, but somehow expected since they are the direct consequence of the reward structure of the game.

Weaknesses

- The paper is not self-contained. Key parts of the paper are in the Appendix. It is not possible to understand the dynamics of the game without checking the Appendix. Several key experimental results are in the Appendix.
- It is difficult to see a clear scientific contribution in this work. The reviewer understands that the authors are making available a new environment for the community, but
- Most of the description of the technical details of the implementation of Lux and its evaluation are in the Appendix. It is worth noting that the some of the notation itself cannot be found in the main paper.
- The actual contribution of this work is unclear, since it mixes the presentation of the platform and the evaluation with a specific curriculum learning algorithm.
- The curriculum learning solution used by the authors is rather standard. The results are not particularly insightful.


**Summary Of The Paper:**

In this paper, the authors present Lux, an environment for multi-agent reinforcement learning at scale. The environment is evaluated through simulations using a curriculum learning solution. The main contribution of this work resides in the environment itself, which allows for a simulation of multi-agent interactions at scale. However, the dynamics of the game itself does not appear particularly insightful. The implementation of the game itself is also rather standard.

**Summary Of The Review:**


The authors present an interesting environment, which is made available to the research community as open source software. However, the environment itself is not particularly innovative compared to the state of the art. The evaluation of the platform considering a curriculum algorithm solution is rather standard. The paper is not well structured and impossible to understand without reading the Appendix. Key sections are not in the main paper.

The reviewer also would like to stress the fact that the paper does not appear to be fully anonymized. There is a direct citation of the GitHub page of the project with the names of the authors of the manuscript.

---

### Official Review · Reviewer_Sp1k · 2022-10-24

**Confidence:** 3
**Correctness:** 3
**Technical Novelty And Significance:** 3
**Empirical Novelty And Significance:** 3
**Recommendation:** 6

**Clarity, Quality, Novelty And Reproducibility:**

The paper is clearly written and is easy to read. The proposed environment is a novel contribution and there are a number of ablation studies performed to study the various strategies that are evolved throughout the learning.

**Strength And Weaknesses:**

The proposed environment captures the essential elements of learning diverse social behaviors required to sustain in a society with limited resources.

One concern I have with the work is the lack of comparison to prior frameworks. I'm skeptical as to what additional features Lux provides that are not already there in previous massive-agent frameworks like MAgent. The comparisons done in the paper do not highlight the main advantage of using Lux over previous frameworks.

**Summary Of The Paper:**

The paper proposes a new benchmark to train a large number of agents in an environment with limited available resources. It also explores the connection of learning atomic / low-level skills with that of learning social strategies such as cooperation, coordination, and competition.

**Summary Of The Review:**

The paper proposes a new massive-agent RL framework that could act as a benchmark for but testing agents' ability to cooperate, coordinate or learn different kinds of social behaviors.

---

### Official Review · Reviewer_ndgW · 2022-10-25

**Confidence:** 4
**Correctness:** 2
**Technical Novelty And Significance:** 2
**Empirical Novelty And Significance:** 3
**Recommendation:** 5

**Clarity, Quality, Novelty And Reproducibility:**

As mentioned in the weaknesses above, it is very important that the authors clarify their role relative to the existing competition. If this is not a double-blind submission, and the nature of the competition that has already occurred would make anonymity impossible, it may be better suited to a single-blind venue such as a Datasets and Benchmarks track (regularly at NeurIPS) or similar. By itself, the algorithmic contributions are not clear and limited to the design of the curriculum training approach, which appears to be very successful. Is the main result that curriculum training is very important for massively multi-agent settings? If so, then this should be emphasized, along with a study of how this approach is useful when paired with other algorithms/approaches. Or is the architecture that is proposed here better-suited for curriculum training specifically? What are the limitations of the centralized policy adopted in the proposed solution?

Minor point: there is a missing reference in line 4 of the Introduction.

**Strength And Weaknesses:**

Strengths:
- This paper demonstrates multi-agent reinforcement learning in a complex domain with limited resources.
- The paper is written clearly and the appendix has many details aiding reproducibility.
- The Lux environment is very compelling and could definitely be a source of important new results in multi-agent reinforcement learning, especially in the context of limited resources.


Weaknesses:
- One of the biggest weaknesses is the nature of the first contribution (introducing the environment itself). This paper mentions introducing this environment, but then cites the Doerschuk-Tiberi and Tao (2021) GitHub page and associated competition. If this paper is coming from the authors of the repo, then this is clearly breaking anonymity. Otherwise, it is not clear why the authors are stating this as a contribution.
- The paper does not adequately explain why the algorithm that is used here has an "up to 90% win rate" against the Isaiah et al. (2021) policy; it appears that this is entirely due to the curriculum training, as mentioned in the caption to Figure 10; otherwise the win rate is only 20%. If that is the case, is the contribution about the effectiveness of curriculum training? If so, how would curriculum training benefit another algorithm in this environment, such as the Isaiah et al. (2021) policy?
- While using a central policy for all of the agents in the solution makes sense from a practical perspective, it primarily makes the action space enormous but removes the individual independence of agents. Would it be possible to train individual agents in this framework, and what would this look like?
- The particular methodology described in Section 4 for a particular neural architecture does not make it clear how difficult it would be to train different neural architectures or to use different algorithms with this framework.
- The accompanying video is not sufficiently self-contained; some text in the appendix explaining what the video demonstrates would be useful.

**Summary Of The Paper:**

This paper introduces a massively multi-agent (thousands of agents) learning environment, Lux, in which teams compete in a custom Real-Time Strategy (RTS) grid world with limited resources. There are two types of agents on each team, Workers and CityTiles, with distinct actions available. The main contributions of the paper, as indicated by the authors, are (1) the introduction of the Lux learning environment; (2) evidence that there are emergent collective behaviors including local strategies such as regional coordination and global strategies such as sustainable development; (3) the implementation details of a particular algorithm using curriculum training that appears to have a very high success rate against the winner of a competition in developing agents for this environment.

**Summary Of The Review:**

This paper needs to clarify what the contributions are (environment vs. curriculum training). If it is the environment, then this should be either submitted to a different venue or anonymized properly before submission. Also, there are a number of questions about what it would take to train agents in this environment, presumably already described in the competition rules referenced in the Appendix. If it is the curriculum training approach, then there needs to be further study of the effect that this has relative to "the base algorithm" it is used with.

---

### Official Review · Reviewer_Qnaj · 2022-10-28

**Confidence:** 3
**Clarity, Quality, Novelty And Reproducibility:** TBD
**Correctness:** 1
**Technical Novelty And Significance:** 1
**Empirical Novelty And Significance:** Not applicable
**Recommendation:** 1

**Strength And Weaknesses:**

TBD

**Summary Of The Paper:**

The paper seems to be deanonymized. "To this end, we propose Lux (Doerschuk-Tiberi & Tao, 2021)" which links to this github repo: "https://github.com/Lux-AI-Challenge/Lux-Design-2021". If this is not the case, I will provide a full review.

**Summary Of The Review:**

TBD

---

### Decision · Program_Chairs · 2023-01-20

**Decision:**

Reject

**Justification For Why Not Higher Score:**

Unanimous agreement among reviewers.

**Justification For Why Not Lower Score:**

N/A

**Metareview: Summary, Strengths And Weaknesses:**

I thank the authors for their submission and active engagement during the discussion period. The reviewers unanimously agree that this paper is not ready for publication. In particular, the reviewers remarked missing clarity around the contributions and a lack of comparison to similar frameworks. Therefore I recommend rejection.